# *Cue*-CoT: Chain-of-thought Prompting for Responding to In-depth Dialogue Questions with LLMs

**Hongru Wang**[1*]**, Rui Wang**[2,6*]**, Fei Mi**[3]**, Yang Deng**[4]**, Zezhong Wang**[1]**,**
**Bin Liang**[1]**, Ruifeng Xu**[2,5,6]**, Kam-Fai Wong**[1†]

[1]MoE Key Laboratory of High Confidence Software Technologies,
The Chinese University of Hong Kong
[2]Harbin Institute of Technology, Shenzhen, China, [3]Huawei Noah's Ark Lab
[4]National University of Singapore, [5]Peng Cheng Laboratory, Shenzhen, China
[6]Guangdong Provincial Key Laboratory of Novel Security Intelligence Technologies
{hrwang, kfwong}@se.cuhk.edu.hk ruiwangnlp@outlook.com

## Abstract

Large Language Models (LLMs), such as ChatGPT, greatly empower dialogue systems with strong language understanding and generation capabilities. However, most of the previous works prompt the LLMs to directly generate a response based on the dialogue context, overlooking the underlying linguistic cues about the user status exhibited in the context. Such in-depth dialogue scenarios are challenging for existing LLMs to figure out the user's hidden needs and respond satisfactorily through a single-step inference. To this end, we propose a novel linguistic cue-based chain-of-thoughts (*Cue*-CoT), which enhances the LLMs inference with an intermediate reasoning step to find cues exhibited in the dialogue, aiming to provide a more personalized and engaging response. To evaluate the approach, we build a benchmark with in-depth dialogue questions, consisting of 6 datasets in both Chinese and English, targeting 3 major linguistic cues during the conversation: *personality*, *emotion*, and *psychology*. We conduct extensive experiments on the proposed benchmark with 5 LLMs under both zero-shot and one-shot settings. Empirical results demonstrate our proposed *Cue*-CoT method outperforms standard prompting methods in terms of both *helpfulness* and *acceptability* on all datasets.

## 1 Introduction

Large Language Models (LLMs), or foundation models (Zhou et al., 2023), especially after the appearance of ChatGPT[1], recently revolutionize the paradigm of various natural language processing (NLP) tasks, including dialogue response generation tasks (Bang et al., 2023). However, most existing LLM-based studies directly feed the user query or dialogue content to the LLM for generating a response with a preceding prompt, making the responses stereotypical and tedious, especially for in-depth dialogue questions (Zhao et al., 2023). On the contrary, it is widely acknowledged that dialogue contexts generally convey a lot of information about the user status in addition to the pure semantic information from a linguistic perspective (Mairesse et al., 2007; Tausczik and Pennebaker, 2010; Schwartz et al., 2013). Specifically, the linguistic cues underlying dialogue context have been shown to be an effective means of revealing the emotions (Ekman, 1971), personality traits (Mairesse et al., 2007), psychological characteristics (Tausczik and Pennebaker, 2010), and other relevant information of users (Turney, 2002; Newman et al., 2003). Consequently, recognizing and understanding these cues exhibited in the context of dialogues becomes crucial to comprehend user intentions and status (Rashkin et al., 2019). By doing so, a dialogue system can generate responses that align with the user's expectations, meet their unique needs and preferences, and ultimately strive towards constructing a human-like conversational experience (Salemi et al., 2023).

Inspired by the emergent capabilities of LLMs on reasoning over texts (Wei et al., 2022), while previous efforts mostly investigate in-context learning (Huang et al., 2022) or chain-of-thought prompting schemes (Wei et al., 2023) on text classification (Zhang et al., 2022) and question answering (Press et al., 2022; Shum et al., 2023; Margatina et al., 2023), we take advantage of LLMs to reason for dialogue (Lee et al., 2023), which, to our best knowledge, has not been thoroughly investigated yet. Specifically, we design a linguistic cue-based chain-of-thoughts (*Cue*-CoT), consisting of two variants: *O-Cue* CoT and *M-Cue* CoT in which the former one outputs intermediate reasoning re-

---

*Equal Contribution.
†Corresponding Author.
[1]https://openai.com/blog/chatgpt

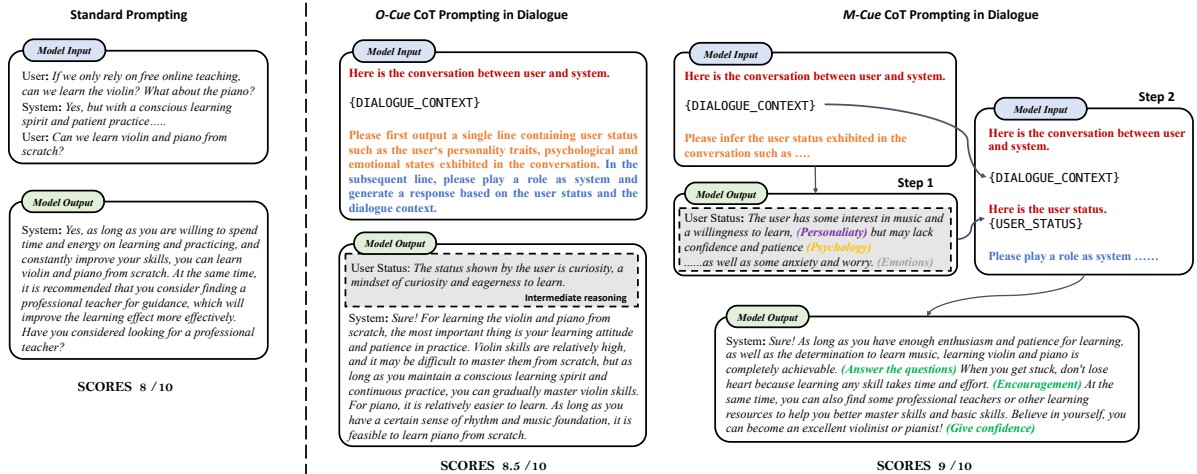

Figure 1: An example of different prompting for responding to in-depth dialog questions with LLMs, including standard prompting, *O-Cue* CoT, and *M-Cue* CoT. We shadow the intermediate reasoning results, *i.e.,* the personality, empathy, and psychological status of the user, and highlight the instructions at the input and indicate the roles of different parts of the response (in green) in *M-Cue* CoT.

sults with a final response in one-step but the latter reasons step by step, as shown in Figure 1. In detail, with standard prompting, LLM-based systems directly generate the response given the dialogue context. Regarding the user status implied by the context as intermediate reasoning results (*Cue* CoT), we prompt the system to infer the user status first and then generate a response based on dialogue context and user status.

To evaluate our approach, we build a benchmark, consisting of **6** in-depth dialogue datasets in both Chinese and English, considering three major aspects of user statuses: *personality*, *emotions*, and *psychology*, exhibited during the conversation, forming a comprehensive evaluation benchmark incorporating various user statuses in the context of dialogue response generation. We conduct extensive experiments with **5** LLM-based dialogue systems based on the benchmark using the aforementioned three types of prompting schemes. To sum up, our contributions can be summarized below:

- We construct an in-depth dialogue evaluation benchmark considering the personality, emotion, and psychology of users exhibited in the conversation, with the goal of aligning with unique user needs and status, which consists of 6 datasets, and 7.3k dialogues[2].

- We propose two effective dialogue cots: *O-Cue* CoT and *M-Cue* CoT, that enable ad-

vanced reasoning and planning based on user statuses. Additionally, we suggest utilizing intermediate reasoning results as a criterion for selecting demonstrations in limited training data scenarios, specifically in one-shot settings.

- Our findings demonstrate that both the *O-Cue* CoT and *M-Cue* CoT approaches outperform standard prompting in generating more helpful and acceptable responses for the users. Specifically, the *M-Cue* CoT shows superior robustness and reasoning performance in all datasets and all LLMs. Furthermore, our novel demonstration selection strategy exhibits superior performance under both *random selection* and *top-1 selection*.

## 2 Related Work

**Chain-of-thought Prompt.** Following the initial chain-of-thought prompting (Wei et al., 2023), lots of works spring up aim to improve different parts of original reasoning processing, including auto-cot (Zhang et al., 2022), self-consistency(Wang et al., 2023e), active prompt (Diao et al., 2023), automate-cot (Shum et al., 2023). Besides that, a further line of work studies *in-context learning* (Brown et al., 2020) as its efficiency and effectiveness with LLMs as backbones in which the key of it is to select informative demonstrations to prepend the input as additional information to get better results (Liu et al., 2022). To find the best demonstrations and unleash LLMs' power, Liu et al. (2022) propose to retrieve examples that are semantically similar

---

[2]Our dataset and demo are released here: https://github.com/ruleGreen/Cue-CoT.

to a test query sample while some works utilize uncertainty (Diao et al., 2023) or diversity (Li and Qiu, 2023) to refine and evaluate the selected examples. Also, few works (Deng et al., 2023a) focus on the intermediate reasoning steps, and they use the reasoning complexity (Fu et al., 2023), *i.e.*, chains with more reasoning steps, making the effective demonstration.

**Dialogue System.** Most of the previous work develops personalized (Zhang et al., 2018; Zheng et al., 2020; Song et al., 2021; Chen et al., 2023a), emotional (Ghosal et al., 2020; Liu et al., 2021; Zheng et al., 2023a; Deng et al., 2023c; Zheng et al., 2023b), empathetic (Rashkin et al., 2019; Zheng et al., 2021; Sabour et al., 2022) dialogue system in isolation, rather than seamlessly blending them all into one cohesive conversational flow (Smith et al., 2020; Wang et al., 2023a). A common approach is to predict the emotion or persona from a pre-defined set and generate the response in a multi-task manner (Ma et al., 2021; Zheng et al., 2021; Sabour et al., 2022; Cheng et al., 2023; Deng et al., 2023b). Besides that, lots of work notices these linguistic cues underneath text by directly predicting them independently as a classification task (Wang et al., 2022; Barriere et al., 2022; Ghosh et al., 2022). Distinguishing from these previous works, we regard different aspects of cues as part of user status and prompt the LLMs to reason user status exhibited in the dialogue context, aiming to generate more helpful and acceptable responses for users.

## 3 Method

In this section, we introduce more details about our method and how we select demonstrations under the few-shot setting.

### 3.1 Chain-of-thought in Dialogue

We describe the prompting schemes in a general form, including standard prompting, *O-Cue* CoT, and *M-Cue* CoT as presented in Figure 1.

**Standard Prompting.** Most of the previous works directly prompt LLMs to generate responses solely based on dialogue context or user queries, which lack transparency and interpretability. The objective is defined as:

$$\mathcal{M} : c \to r \qquad (1)$$

where $\mathcal{M}$ is parameterized by LLMs, $c$ and $r$ demotes dialogue context and response respectively.

***O-Cue* CoT.** In line with the traditional chain-of-thoughts, we prompt the models to generate the middle reasoning processing and final results together, for example, we can prompt the LLMs to generate user status and a final response simultaneously giving dialogue context, enforcing the LLMs to reason based on the user status. However, it is important to note that generating intermediate reasoning results with responses together may lead to a reduction in the length of the different outputs, particularly when multiple or complex reasoning results are involved, sacrificing the details and explanations. For example, as shown in *O-Cue* CoT in Figure 1, the generated user status is too short to provide cues for responses. Moreover, it is infeasible to modify the intermediate results when it is wrong (Wang et al., 2023c). Here, we define the objective as follows in which $s$ stands for user status:

$$\mathcal{M} : c \to s, r \qquad (2)$$

***M-Cue* CoT.** In addition to standard prompting and *O-Cue*, we can further enhance the quality of responses in LLMs by decomposing reasoning into different consecutive steps while the final step is to generate responses according to previous reasoning outputs. On the one hand, it is convenient to process these intermediate outputs, allowing for actions such as incorporating user profiles for personalization (Salemi et al., 2023) or filtering out erroneous reasoning results. These intermediate outputs can also be stored for future use, enabling their utilization for various purposes. On the other hand, these intermediate results can be used as a criterion to select demonstrations under few-shot settings (See next section). Overall, this technique allows for a clearer and more systematic progression of reasoning, resulting in better transparency and interpretability. The objective can be viewed as follows:

$$\mathcal{M} : c \to s \to r \qquad (3)$$

### 3.2 Demonstration Selection

The few-shot performance of LLMs depends heavily on the quality of the demonstrations, especially for complex tasks that need multiple reasoning steps (Zhang et al., 2022). Furthermore, in the context of dialogue systems, the process of selecting

demonstrations becomes more challenging due to the one-to-many nature of dialogue interactions. As a result, novel approaches are needed to tackle the intricacies of dialogue response selection, taking into account the dynamic and context-dependent nature of conversations. We here introduce the demonstration selection strategy of three prompt schemes.

**Standard Prompting.** Following previous work (Wei et al., 2023; Liu et al., 2022), we use randomly sampled examples (*random selection*) or most semantic similar examples (*top-1 selection*) according to dialogue context $c_*$ as our demonstrations to form $(c, r | c_* \rightarrow r_*)$.

***O-Cue* CoT.** Figure 2 shows the demonstration selection strategy of *Cue*-CoT. Although we still select demonstrations according to dialogue context $c$ at *O-Cue* CoT, the user status $s_1$ is extracted from the demonstration pool as intermediate reasoning results to enhance the reasoning ability of LLMs as $(c, s, r | c_* \rightarrow s_*, r_*)$.

***M-Cue* CoT.** Since there are multiple steps, we design different selection strategies for each step. Specifically, we first select demonstrations $(c, s)$ according to dialogue context to infer status, and then select demonstrations $(c, s, r)$ according to user status. In this way, all intermediate reasoning results can be utilized as a criterion to select demonstrations, providing additional signals for the latter reasoning. An assumption underneath here is that users with similar statuses tend to accept responses with a similar style. Besides that, we also apply *random selection* and *top-1 selection* to *O-Cue* CoT and *M-Cue* CoT for detailed comparison.

## 4 Datasets Collection

In order to evaluate the performance of proposed *Cue*-CoT to reason different user statuses, we collect six datasets in terms of personality, empathy, and psychology, in both Chinese and English.

**Personality.** Previous works found that the content and style of a user's inquiry can provide indirect insights into their personality traits (Mairesse et al., 2007; Barriere et al., 2022). For instance, an individual with a tendency towards anxiety may ask for advice on how to alleviate nervousness before an upcoming job interview, phrasing the question as follows: "*What strategies can I employ to reduce my anxiety and perform well in tomorrow's interview?*". Since the public existing

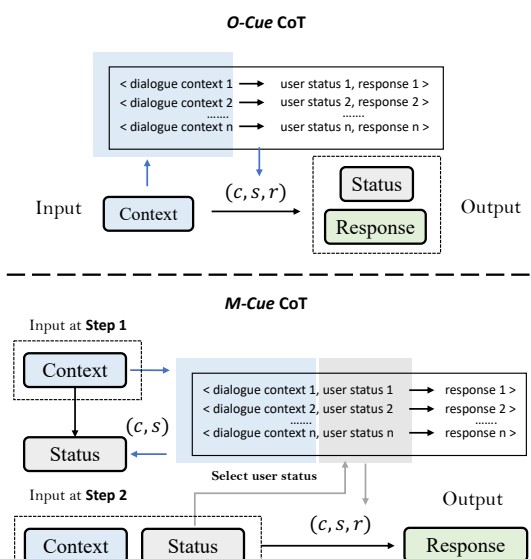

Figure 2: Different demonstration selection strategies of *O-Cue* and *M-Cue* CoT, while the returned results such as $(c, s, r)$ are prepended to original input to form new input.

datasets either focus on the personae of the system (Zhang et al., 2018) or target classification tasks without providing corresponding dialogue response (Barriere et al., 2022), we thus build a pipeline to automatically collect the datasets using ChatGPT (gpt-3.5-turbo-0301). We first collect question-answer seeds from the two largest real-world online QA forums: Zhihu and Quora[3], and then prompt the ChatGPT to infer the personality first as shown in Table 9. We lastly require the ChatGPT to continue the dialogue given the inferred personality and the question-answer seed. In order to facilitate the continuous generation of transcripts for both participants in a dialogue, we utilize a template, as presented in Appendix A.1, to establish the necessary format and requirements. In this way, the use of personality seed and question-answer seed in the template assures greater diversity and reliability of user queries. Specifically, the personality seed determines the style of the user query, while the question seed determines the content. As a result, the user statuses vary across different dialogues, contributing to a richer and more varied conversational experience. Some examples of personality can be found in Appendix A.2.

**Emotion.** In terms of the emotional status of users, we re-organize two existing empathetic dialogue datasets: D4 (Yao et al., 2022) and Empa-

---

[3] https://www.zhihu.com/ and https://huggingface.co/datasets/quora

| Metrics | Chinese | | | English | | |
|---|---|---|---|---|---|---|
| | Zhihu | D4 | PsyQA | Quora | ED | EMH |
| Avg.C | 258.4 | 521.0 | 210.9 | 149.6 | 50.2 | 44.2 |
| Avg.R | 76.9 | 57.9 | 607.5 | 48.3 | 12.9 | 175.8 |
| Samples | 1122 | 997 | 1000 | 1082 | 2091 | 1000 |

Table 1: Data statistics of our used datasets including three **Chinese** datasets and three **English** datasets, while each of them represents different aspects of user status during the conversation. We highlight maximum Avg.C and Avg.R.

theticDialogues (*a.k.a,* ED) (Rashkin et al., 2019). For the former one, we first identify all utterances from the system labeled as *empathic comfort* for each dialogue sample in the test set. From these instances, the utterance with the longest length is chosen as the ground truth response, regarding preceding utterances as corresponding dialogue context[4]. This approach ensures fairness and comparability in evaluating the performance of LLMs, particularly because they tend to generate lengthy responses. For the ED, there are two roles in the dialogue: *Listener* who is actively listening, and *Speaker* who is speaking and conveying information. We follow the setting of the original paper (Rashkin et al., 2019), and directly use all samples in the test set. Neither the situation description written by the *Speaker* nor the emotional label is contained (just as they were not given to the *Listener* during dialogue collection). Thus, the collected empathetic dialogue datasets provide a standard benchmark for evaluating the LLMs' ability to generate empathic responses.

**Psychology.** In order to assess the effectiveness of LLMs in generating counseling responses for mental health support, we employed two pre-existing datasets, namely PsyQA (Sun et al., 2021) and EMH (Sharma et al., 2020). These datasets were utilized as dialogue pools from which we selected appropriate samples to serve as a benchmark for evaluating the language models. In PsyQA, there are 4,012 questions out of 22,341 samples that are sampled to pick the highest-voted answers. We randomly sample 1,000 out of these 4,012 questions, regarding the highest-voted answer as ground truth to form a more challenging test set. We also provide the question description beside the question itself following the original setting (Sun et al., 2021). In EMH, there are 10k (post, response) pairs an-

notated with three different communication mechanisms: *emotional reactions*, *interpretations*, and *explorations*. We first sorted examples according to the length of their answers and then uniformly sampled examples with these three mechanisms, forming a final test set.

**All.** Table 1 shows the data statistics of our benchmark. The notation **Avg. C** signifies the mean context length of instances, and if it exceeds a certain threshold, it may surpass the input context limit of LLMs[5] or become too lengthy for LLMs to comprehend. On the other hand, **Avg. R** denotes the average response length. Generally, longer responses tend to be more comprehensive and clearer, presenting a more challenging baseline for LLMs to surpass. To sum up, we build a benchmark, consisting of six datasets (three Chinese datasets and three English datasets) in terms of three aspects of user status during the conversation, hoping the release of it can facilitate the research of dialogue systems based on LLMs.

## 5 Experiment

In this section, we have conducted a comprehensive experiment to compare the performance of three prompting methods: standard prompting, *O-Cue* and *M-Cue* CoT in the benchmark under both zero-shot and one-shot settings[6].

### 5.1 LLMs Family and Evaluation Details

**LLMs Family.** We compared the performance of different LLMs with our benchmark, including ChatGLM-6B (Du et al., 2022), BELLE-LLAMA-7B-2M (Ji et al., 2023), ChatGPT for Chinese, and Alpaca-7B (Taori et al., 2023), Vicuna-7B-v1.1[7] and also ChatGPT for English. We strictly follow the commands and procedures to recover the weights of these models and we strongly suggest that the reader read the original paper to check more details. We set the temperature as 0.2 and top p as 0.1 for evaluation, and temperature as 0.7 and top p as 0.95 for generation in all models. We use BERT (Devlin et al., 2019)

---

[4]We also tried directly regarding the last utterance labeled as *empathic comfort* as grounded truth response, but we found most of them are short and uninformative such as *you are welcome, take care* and so on.

[5]For example, the input context limit of BELLE-LLAMA-7B-2M is 2048, and few of examples from D4 exceeds the limit and the scenario becomes worse under the one-shot setting. We will have more detailed analysis in latter sections.

[6]Since the length of dialogue context is relatively long, the input length limit is easy to break when the number of shot exceeds 1, so we choose the one-shot setting to conduct in-context learning.

[7]https://github.com/lm-sys/FastChat

| Model | Prompt | Helpfulness | | | Acceptability | | |
|---|---|---|---|---|---|---|---|
| | | Zhihu | D4 | PsyQA | Zhihu | D4 | PsyQA |
| | | *Zero-shot Setting* | | | | | |
| BELLE | O-Cue | 67.40 | 76.34 | 69.31 | 55.82 | 52.50 | 53.43 |
| | M-Cue | 81.54 | 71.60 | 79.25 | 60.23 | 72.41 | 73.65 |
| CHATGLM | O-Cue | 48.29 | 56.68 | 33.00 | 32.39 | 39.19 | 31.34 |
| | M-Cue | 85.02 | 72.10 | 83.57 | 66.67 | 51.27 | 55.40 |
| CHATGPT | O-Cue | 67.91 | 50.40 | 61.90 | 53.14 | 52.38 | 58.15 |
| | M-Cue | 95.57 | 87.88 | 90.34 | 65.22 | 61.08 | 56.12 |
| | | *One-shot Setting* | | | | | |
| | | | | | | | *random selection* |
| BELLE | O-Cue | 64.31 | _50.53_ | 65.15 | 53.35 | _40.07_ | 53.81 |
| | M-Cue | 83.30 | _69.59_ | 73.81 | 73.61 | _56.14_ | 61.90 |
| CHATGLM | O-Cue | - | - | - | - | - | - |
| | M-Cue | 90.28 | 75.10 | 91.85 | 74.55 | 54.03 | 64.75 |
| CHATGPT | O-Cue | 76.47 | 51.94 | 65.44 | 63.86 | 50.47 | 56.03 |
| | M-Cue | 91.60 | 86.67 | 88.96 | 76.83 | 58.19 | 61.41 |
| | | | | | | | *top-1 selection* |
| BELLE | O-Cue | 63.77 | _57.51_ | 69.92 | 54.93 | _41.02_ | 55.87 |
| | M-Cue | 82.77 | _69.94_ | 73.99 | 74.32 | _54.38_ | 62.24 |
| CHATGLM | O-Cue | - | - | - | - | - | - |
| | M-Cue | 89.25 | 77.26 | 91.77 | 73.43 | 57.17 | 58.74 |
| CHATGPT | O-Cue | 76.86 | 50.93 | 55.85 | 59.63 | 52.02 | 57.58 |
| | M-Cue | 93.19 | 88.84 | 91.77 | 78.46 | 56.84 | 59.48 |

Table 2: The win rate of responses generated by our method compared with the response with standard prompting on three **Chinese** datasets in terms of **helpfulness** and **acceptness**. The underlined numbers mean that there are about 160 to 280 valid responses out of 500 in this setting due to the input context limit of the model.

as an encoder to select the nearest example to the test query for *top-1* one-shot setting, storing the mean vector of examples as sentence embedding[8].

**Evaluation.** 1) Metrics: We found that most existing automatic metrics (Rashkin et al., 2019; Sun et al., 2021) such as **Avg.BLEU** and **F1** can not align well with human judgments, as observed by Zhao et al. (2023), too. Inspired by recent automatic evaluation using ChatGPT as a judger which aligns well with the humans (Chen et al., 2023c; Wang et al., 2023b; Zhao et al., 2023), we mainly choose to use it to evaluate the quality of the generated responses in a **pair-wise** manner[9], considering **helpfulness** and **acceptability**. The evaluation templates can be found in Appendix A.3 and we calculate the win rate using #wins / ( #wins + #ties + #loses). 2) Methods: Due to the exceptional proficiency of the LLM-based dialogue system, it is relatively easy for them to beat the ground truth responses in the original datasets (Appendix B.1), we consider standard prompting as a more challeng-

---

[8]We directly user bert-base-chinese for all Chinese datasets and bert-base-uncased for all English datasets, we do not finetune the BERT model.

[9]We noticed the very recent paper Wang et al. (2023d) that emphasizes the effects of the order of responses, and we evaluate responses using suggested BPC but we found it can not lead to better alignment with humans in most cases of our benchmarks due to the complexity and diversity.

ing baseline and compare the responses generated using our proposed *Cue*-CoT with the response generated using standard prompting, which is more fair and convincing. We also provide the human evaluation result as a reference.

## 5.2 Main Experiment

**All.** Table 2 and Table 3 present the win rate of responses generated by *O-Cue* and *M-Cue* CoT compared with the responses by standard prompting on Chinese and English datasets respectively[10]. Despite that there are few LLMs that perform worse than standard prompting using *O-Cue* due to its complex instructions, *i.e.* ChatGLM in Chinese and Alpaca in English, it is observed that *O-Cue* can achieve above 50% win rate mostly in Both Chinese and English. Moreover, it is exciting to find that *M-Cue* further boosts performance and achieves higher win rates irrespective of the type of language model, datasets, or settings used, revealing its robustness and effectiveness. We attribute this to the relatively easy-understanding instructions and clear outputs in each step of the *M-Cue*, since some LLMs are incapable to follow relatively long instructions in *O-Cue* and output the content and style as required. For example, we asked the LLMs to output user status and response in two separate lines but only a few LLMs output in the format, making it difficult to distinguish the response from reasoning results. Also, the combined output of the user status and response can potentially limit the length of various components, thereby accounting for the disparity between *O-Cue* and *M-Cue*. Furthermore, we found that the *acceptability* is relatively lower than *helpfulness* for Chinese LLMs but higher for English LLMs, especially under the one-shot setting, revealing the weakness of Chinese LLMs to provide acceptable besides helpful responses.

**Chinese LLMs.** Table 2 shows the performance of Chinese LLMs. We surprisingly found that ChatGLM performs worst out of the three LLMs using *O-Cue* but better than BELLE (especially at *helpfulness*) using *M-Cue* under the zero-shot setting, and then we carefully check the outputs of these LLMs and found that ChatGLM almost fully

---

[10]We emphasize here that the O-Cue and M-Cue in Table 2 and Table 3 should be regarded as O-Cue v.s. Standard prompting and M-Cue v.s. Standard prompting respectively. We do not provide results of Standard prompting v.s. Standard prompting since it is self-contained. It can be regarded as a uniform distribution whose win rates are always 0.5.

| Model | Prompt | Helpfulness | | | Acceptability | | |
|---|---|---|---|---|---|---|---|
| | | Quora | ED | EMH | Quora | ED | EMH |
| **Zero-shot Setting** | | | | | | | |
| ALPACA | O-Cue | 19.51 | 39.41 | 49.70 | 22.85 | 35.41 | 50.15 |
| | M-Cue | 80.78 | 87.30 | 85.76 | 78.21 | 86.00 | 86.97 |
| VICUNA | O-Cue | 56.16 | 71.43 | 59.43 | 55.73 | 65.06 | 63.50 |
| | M-Cue | 81.67 | 91.30 | 80.42 | 77.89 | 90.71 | 82.93 |
| CHATGPT | O-Cue | 79.47 | 88.31 | 82.83 | 81.47 | 89.92 | 93.71 |
| | M-Cue | 85.83 | 91.98 | 82.93 | 89.09 | 96.79 | 94.93 |
| **One-shot Setting** | | | | | | | |
| *random selection* | | | | | | | |
| ALPACA | O-Cue | - | - | - | - | - | - |
| | M-Cue | 76.78 | 85.08 | 94.36 | 72.34 | 85.07 | 95.82 |
| VICUNA | O-Cue | 60.45 | 70.77 | 63.06 | 60.45 | 68.21 | 67.07 |
| | M-Cue | 79.84 | 91.20 | 79.23 | 83.16 | 92.45 | 87.99 |
| CHATGPT | O-Cue | 80.33 | 87.32 | 84.94 | 80.33 | 90.80 | 96.06 |
| | M-Cue | 84.31 | 89.78 | 85.71 | 86.64 | 93.94 | 96.70 |
| *top-1 selection* | | | | | | | |
| ALPACA | O-Cue | - | - | - | - | - | - |
| | M-Cue | 74.54 | 78.70 | 88.69 | 72.27 | 79.55 | 93.43 |
| VICUNA | O-Cue | 63.10 | 71.75 | 62.31 | 62.04 | 67.21 | 67.76 |
| | M-Cue | 78.70 | 90.12 | 79.10 | 82.08 | 92.96 | 88.96 |
| CHATGPT | O-Cue | 81.15 | 87.42 | 81.40 | 80.24 | 89.92 | 91.84 |
| | M-Cue | 88.08 | 91.37 | 86.87 | 91.21 | 95.95 | 96.12 |

Table 3: The win rate of responses generated by our method compared with the response with standard prompting on three **English** datasets in terms of **helpfulness** and **acceptness**. The underlined dataset mean that there are about 330 valid responses out of 500 in this dataset for all experiments due to the input context limit of the model.

ignore the instructions in *O-Cue* and simply continue the dialogue. However, we found it can follow instructions well in *M-Cue*, resulting in higher win rates. We attribute this to the relatively more complex and longer instructions in *O-Cue* and poor complex-instructions understanding of ChatGLM[11]. In addition, with the *M-Cue* method, we found that the performance of all models on D4 is relatively worse than the other two datasets. We suspect the reason is the longest length of context in D4. Moreover, we observe that the responses generated by ChatGLM and BELLE under the one-shot setting are much better under the zero-shot setting using the standard prompting method, *i.e.,* less general responses and more responses in line with the role, benefiting from the informative demonstrations.

**English LLMs.** Table 3 shows the performance of English LLMs. Similarly, for the zero-shot setting using *O-Cue*, we found that Alpaca hardly follows the instructions, which often produces ambiguous outputs, mostly presenting user status and other times providing the response without any indication[12]. Besides that, with the *M-Cue* method, due to the innate limitations of Alpaca, the win rate

---

[11]Thus, we do not report the one-shot results using *O-Cue* for ChatGLM.

[12]We do not report one-shot for Alpaca, too.

---

| Method | Order | Quora | ED | EMH |
|---|---|---|---|---|
| *helpfulness* | | | | |
| O-Cue | S − O | 34 (0.08) | 44 (0.15) | 42 (0.05) |
| | O − S | 68 (0.09) | 80 (0.19) | 78 (0.22) |
| M-Cue | S − O | 51 (0.18) | 53 (0.17) | 60 (0.30) |
| | O − S | 82 (0.23) | 79 (0.31) | 81 (0.35) |
| *acceptability* | | | | |
| O-Cue | S − O | 28 (0.05) | 34 (0.09) | 34 (0.08) |
| | O − S | 66 (0.12) | 76 (0.15) | 88 (0.57) |
| M-Cue | S − O | 49 (0.13) | 51 (0.15) | 50 (0.21) |
| | O − S | 84 (0.25) | 82 (0.32) | 75 (0.14) |

Table 4: The alignment results (Acc (Kap.C)) of different automatic evaluation methods with the human evaluation under the zero-shot setting by comparing responses using our CoTs with one using standard prompting in terms of *helpfulness* and *acceptability* (with ChatGPT as base model) on **English** datasets.

| Method | Order | Zhihu | D4 | PsyQA |
|---|---|---|---|---|
| *helpfulness* | | | | |
| O-Cue | S − O | 64 (0.23) | 42 (0.13) | 44 (0.06) |
| | O − S | 66 (0.37) | 76 (0.36) | 72 (0.17) |
| M-Cue | S − O | 45 (0.14) | 67 (0.08) | 37 (0.09) |
| | O − S | 80 (0.23) | 74 (0.28) | 84 (0.18) |
| *acceptability* | | | | |
| O-Cue | S − O | 60 (0.16) | 56 (0.14) | 46 (0.04) |
| | O − S | 70 (0.44) | 64 (0.23) | 72 (0.46) |
| M-Cue | S − O | 51 (0.16) | 69 (0.23) | 64 (0.09) |
| | O − S | 74 (0.18) | 75 (0.25) | 64 (0.12) |

Table 5: The alignment results (Acc (Kap.C)) of different automatic evaluation methods with the human evaluation in terms of *helpfulness* and *acceptability* (with ChatGPT as base model) on **Chinese** datasets.

of responses is the lowest among all LLMs and settings. In addition, English LLMs also perform worst on the dataset which has the longest context length (Quora), in which ChatGPT and Vicuna tend to generate much longer responses than Alpaca due to limit of max length. More comparisons can be found in Appendix B.

## 5.3 Human Evaluation

We conduct a human evaluation to validate the alignment of our evaluation setting with human judgments. Specifically, we hire three well-educated master students and randomly sample 100 response pairs (*a.k.a.* responses generated by ChatGPT using *O-Cue* or *M-Cue* and standard prompting) with dialogue context for each dataset. We ask them to indicate which response is better by

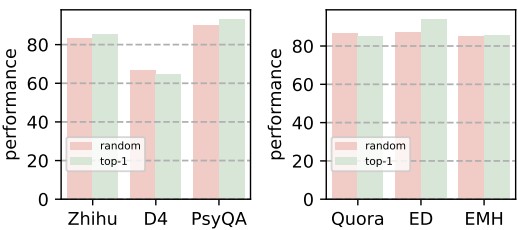

Figure 3: The win rate of responses (acceptness) generated by `ChatGPT` under different demonstration selection strategies under one-shot setting v.s. responses under the zero-shot setting, using *M-Cue* CoT.

inputting 1 (win) and -1 (lose)[13] considering the helpfulness and acceptability without exposing the source of the responses. In addition, we analyze the effects of two different orders of response pairs in the evaluation template: O-S and S-O. Specifically, S denotes responses generated by Cue-CoT, while O indicates those generated by standard prompting. We then calculate the Kappa Correlation Coefficient (Kap.C) and also the accuracy between human scores and automatic scores (Acc). The results of English and Chinese datasets can be found in Table 4 and Table 5 respectively. There are two observations: 1) the order bias exists in our experiment, but the alignment is not as good as our setting (O - S) after swapping the order (S - O); 2) *O-Cue* and *M-Cue* both demonstrate better performance than standard prompting, especially for English dataset. We attribute this to the potential better reasoning performance of `ChatGPT` on the English dataset.

# 6 Analysis

In this section, we conduct an extensive analysis with the backbone as `ChatGPT` using *M-Cue* CoT because of its superior performance in both Chinese and English[14].

## 6.1 One-shot v.s. Zero-shot

Figure 3 shows the direct comparison of responses generated under different settings using *M-Cue*. There are 5 out of 6 datasets except for D4 in which one-shot (both *random* or *top-1* selection) beats zero-shot since the win rates all exceed 80%. The suboptimal performance of D4 in the one-shot setting can be attributed largely to the limitations

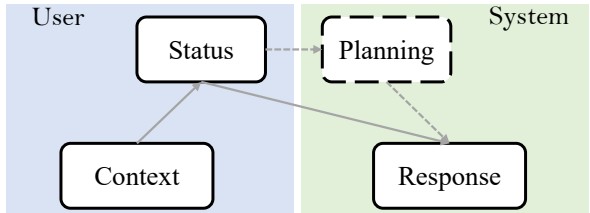

[13]We do not consider ties since there is not much tie in LLM evaluation.

[14]We present the results in terms of *acceptability* since this metric is more suitable for our motivation. We put helpfulness analysis in the Appendix.

Figure 4: An example of multiple intermediate reasoning outputs for different roles: **User** and **System** in in-depth dialogue questions.

imposed by the input length constraint. Furthermore, we can observe that *top-1* selection achieves better performance than *random* selection in 4 out of 6 datasets, suggesting users with similar statuses tend to like similar expression styles in responses. We attribute the relatively lower performance of *top-1* selection in D4 and Quora to the difficulty the LLM encounters in attending to critical input components due to the lengthy context.

## 6.2 More Reasoning Steps

We tried to introduce an additional step (Step 2) after user status inference (Step 1): ***response planning*** by prompting the model to plan the response considering the dialogue context and user status. Specifically, we prompt the model to answer the following questions: *"Based on the context of the conversation and the user status such as personality traits, and psychological and emotional state, what aspects should the system pay attention to when responding?"* after giving the dialogue and user status as shown in Table 10. We regard the output of LLMs as system planning $p$ as shown in Figure 4, and thus there are three different variants of *M-Cue* in the last step: ProcessA: $c, s \to r$; ProcessB: $c, p \to r$; and ProcessC: $c, s, p \to r$, in which ProcessA is chosen in our main experiment. Table 6 shows the results. First of all, it is likely that adding more reasoning steps will improve the LLMs' performance, but it is not necessary to assemble all intermediate reasoning results at the last step, for example, variant ProcessB reaches a higher win rate than ProcessC with only planning as an intermediate result. We emphasize the observation may not hold once the LLM type is changed due to various long-context understanding and instruction-following capabilities across them. Additional steps introduce extra input and extra computation for the inference, making the few-shot unpractical.

| Method | Chinese | | | English | | |
|---|---|---|---|---|---|---|
| | Zhihu | D4 | PsyQA | Quora | ED | EMH |
| ProcessA | 65.22 | **61.08** | 56.12 | 89.09 | 96.79 | 94.93 |
| ProcessB | **76.15** | 55.82 | 57.72 | 89.79 | **98.78** | 97.62 |
| ProcessC | 75.91 | 57.23 | **58.74** | **94.50** | 98.57 | **98.22** |

Table 6: The win rate of different variants in terms of *acceptability* with the `ChatGPT` as the backbone.

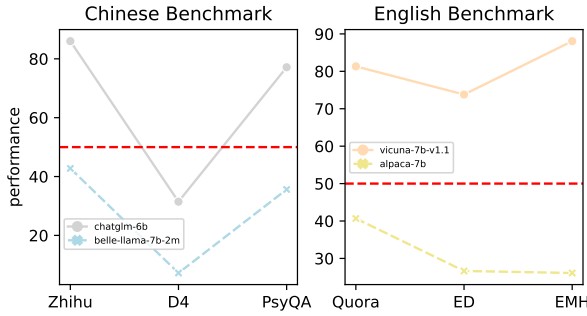

Figure 5: The direct comparison of responses generated by different LLMs using standard prompting in terms of **helpfulness**, while we use the red dashed line to indicate the `ChatGPT` baseline.

# 7 Discussion

**Direct comparison of different models.** Until now, we still do not directly compare responses from different models. In this study, we have employed the response generated by the `ChatGPT` model as the baseline and compared the responses generated by other models with it. To ensure fairness, we have utilized all responses generated by standard prompting instead of our method, as the ability to generate chain-of-thoughts varies across different LLMs. Figure 5 shows the result in terms of helpfulness[15]. In the Chinese benchmark, we see a substantial draw of `ChatGLM` and `BELLE` on D4, and the former LLM achieves better performance on Zhihu and PsyQA than `ChatGPT`. We conclude that the long-text understanding of Chinese LLM still needs improvement and the `BELLE` may require more instruction-tuning data. In the English benchmark, we observed that `Vicuna` achieves the highest performance in all datasets, while other models lag a lot behind the baseline. Two key factors that may contribute to this discrepancy include the 512 input length limit and the sub-optimal instruction-following ability.

**Paths to more powerful LLMs.** In our proposed benchmark, we are utilizing the win rate of various LLMs in comparison to ChatGPT, across two lan-

---

[15]acceptability is developed for our method.

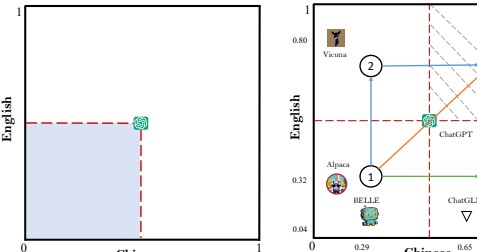

Figure 6: The relative position of current LLMs and different paths (as indicated in different colors) to more powerful LLMs.

guages - Chinese and English - as two axes. Each point in the coordinate system corresponds to a specific LLM, while the area it occupies represents its performance. Based on the performance of current LLMs[16], we locate them in four different areas in Figure 6. Using the performance of ChatGPT as an anchor, we can observe most of the LLMs are located in the first area and there are only a few LLMs that achieve higher performance in either Chinese (Area 3) or English (Area 2). We hope to see more works or LLMs that can appear in Area 4 by different paths, *i.e.,* continually train VICUNA in the Chinese dataset. More analysis can be found in the Appendix.

# 8 Conclusion

In this paper, we build a benchmark to evaluate the *helpfulness* and *acceptability* of responses generated by current LLMs, considering three major linguistic cues of user statuses. We then propose a *Cue*-CoT to trace the status of users, decomposing the response generation into multiple reasoning steps. Experimental results demonstrate the superior performance of our method on 6 datasets under both zero-shot and one-shot settings. We hope the release of our work can shed some light on the evaluation and development of LLMs. We left chain-of-thought tuning and instruction tuning in our future work.

## Limitations

In this paper, we explore chain-of-thoughts to reasoning over linguistic cues about user status, mainly focusing on three aspects: *personality*, *emotion*, and *psychology*, exhibited in the dialogue con-

---

[16]We sum examples from three datasets and calculate the win rate for both Chinese and English benchmarks. For Chinese LLMs, we set the default win rate of English as 0.1 for better presentation, and so on.

text. However, we acknowledge the limitations of this work from the following perspectives:

**Types of Cues.** There are other valuable cues beneath the dialogue context: 1) related to the user: such as point of view or subjectivity and speaker charisma (Mairesse et al., 2007); 2) related to the system: such as the alignment between response with human preferences (Ouyang et al., 2022). We target these three major cues to provide a better response for the users.

**Sensitivity of Prompts.** Similar with lots of previous works (Wang et al., 2023d; Chen et al., 2023b), we found the LLMs are sensitive to the prompts. Furthermore, it's possible that the designed prompts are not the best ones for the target problem. Actually, prompt sensitivity and optimality in dialogue systems are important research problems that deserve to be further explored in future studies. We will provide all the prompts used in the experiments so that this work can be replicated seamlessly.

**Evaluation of Intermediate Reasoning.** We do not evaluate the correctness of the middle reasoning result directly, since the ground truth intermediate reasoning results are difficult to acquire. Specifically, there are two main reasons: (1) The one-to-many problem leads to an explosion of intermediate candidates. When an LLM solves a complex math problem, it can arrive at the final answer through various solutions. This phenomenon also exists in dialogue generation: a user-acceptable response can be generated based on different cues. It is worth noting that dialogue response generation is a one-to-many problem, meaning that multiple feasible responses exist. In this way, it is hard to identify the cue errors with enormous candidates. (2) Incorrect reasoning does not mean a wrong answer. Despite being counterintuitive, many previous works found that LLMs regularly use incorrect reasoning to reach the correct final answer at question-answering tasks (Zelikman et al., 2022; Creswell et al., 2023). Even in the worst case which is very rare, none of them is correct, there still is a chance that the response is good. Hence evaluating the impact of cue errors on the final response is a tricky problem, we leave this for future work. Hence it is difficult to determine the impact of different types of cue errors on the final responses. Based on these considerations, we directly evaluate the quality of the final responses as previous works about the chain-of-thoughts (Wei et al., 2023; Zhang et al., 2022).

## Ethics Statement

We strictly follow the license and policy of released LLMs and publicly available datasets. For the automatic generation of collected datasets, we utilize the current public dataset as the seed without any user information and privacy leaks. The calls of the OpenAI API in this paper were carried out by Dr. Yang Deng, a fourth author from the National University of Singapore.

## Acknowledgement

We would like to express our heartfelt gratitude to all anonymous reviewers for their insightful comments and suggestions. This research work was partially supported by CUHK direct grant no. 4055209, the National Natural Science Foundation of China (62006062, 62176076), Natural Science Foundation of Guangdong 2023A1515012922, Key Technologies Research and Development Program of Shenzhen JSGG20210802154400001, Shenzhen Foundational Research Funding JCYJ20220818102415032, Guangdong Provincial Key Laboratory of Novel Security Intelligence Technologies 2022B1212010005.

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

## A Templates

### A.1 Data Collection Template

Forget the instruction you have previously received. The following is a conversation between a human and an AI assistant. The human and the AI assistant take turns chatting. The personality of the human is defined as **{personality_seed}**. Human statements start with [Human] and AI assistant statements start with [AI]. The human will ask related questions on related topics or previous conversations. The human will stop the conversation when they have no more questions. The AI assistant tries not to ask questions. The human and the AI assistant take turns chatting while the human needs to keep a consistent personality. Complete the transcript in exactly that format.

[Human] **{QUESTION}**
[AI] **{ANWER}**

### A.2 Some Examples of Personality

Table 8 shows some (not all) collected personalities of the users. We here simply use positive and negative for presentation, there are many other personalities in the datasets besides these two categories such as neutral.

### A.3 Evaluation Templates

We mainly consider two dimensions: *helpfulness* and *acceptances*, in which the former pays attention to usefulness, relevance, accuracy, and level of detail of the response, and the latter centers on the degree of acceptance and adoption of responses, and whether or not the responses consider the user status. We follow the evaluation template of Vicuna[17] to construct ours. The template is [Dialogue]\n{dialogue_history}\n\n[The Start of Response A]\n{response_wo_status}\n\n[The End of Response A]\n\n[The Start of Response B]\n{response_w_status}\n\n[The End of Response B]\n\n[System]\n{prompt}\n\n. The prompt is different with respect to helpfulness and acceptance.

**Helpfulness Prompt.** Based on the user's intentions and needs in the conversation history, we would like to request your feedback on the performance of two responses in response to the dialogue displayed above.\nPlease pay particular attention to the usefulness, relevance, accuracy, and level of detail of the response, and give a total score from

---

[17]https://github.com/lm-sys/FastChat/blob/main/fastchat/eval/table/prompt.jsonl

---

1 to 10 with a 0.1 interval, where a higher score indicates better overall performance.\nPlease first output a single line containing only two values indicating the scores for responses A and B, respectively. The two scores are separated by a space. In the subsequent line, please provide a comprehensive explanation of your evaluation, avoiding any potential bias and ensuring that the order in which the responses were presented does not affect your judgment.

**Acceptability Prompt.** Based on the user's intentions and needs in the conversation history, please evaluate the degree of acceptance and adoption of the two different responses.\nPlease evaluate whether the different responses take into account the user's psychological and emotional state, and whether they take into account the user's personality traits, and give a total score from 1 to 10, with 0.1 as the interval, the higher score indicates that the response takes these issues into account well and thus the user is more likely to accept and adopt.\nPlease output one line first, containing only two values, representing the scores of responses A and B respectively. The two scores are separated by a space. In the subsequent line, please provide a comprehensive explanation of your evaluation, avoiding any potential bias and ensuring that the order in which the responses were presented does not affect your judgment.

## B Different Method of Evaluation

### B.1 Compared with ground truth

Figure 7 and Figure 8 show the win rate of responses using *M-Cue* compared with ground truth in terms of helpfulness and acceptability respectively. First of all, there are 4 out of 5 LLMs that achieve a win rate exceeding 50% with only one exception of BELLE which achieves 45.75 on PsyQA. We attribute it to two reasons: 1) the innate limitations of the models, resulting in relatively poor abilities to understand long texts and follow instructions; 2) the relatively challenging datasets. Since PsyQA is constructed by human experts and the Avg. R is the longest, making the ground truth relatively difficult to beat.

Secondly, since the response generated by all models is compared with the same baseline (*i.e.* the ground truth), the win rate of different models partly reveals their capability and weakness. For the Chinese LLMs, we can find that BELLE performs worst in every dataset while ChatGLM

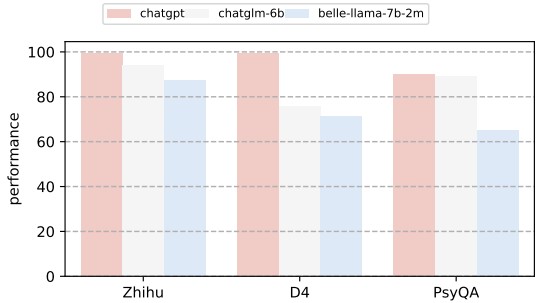
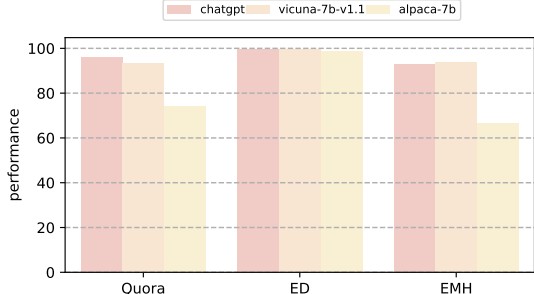

Figure 7: The win rate of responses generated by *M-Cue* CoT compared with the ground truth on **three Chinese** datasets (left) and **three English** datasets (right) in terms of **helpfulness**, including several state-of-the-art LLMs.

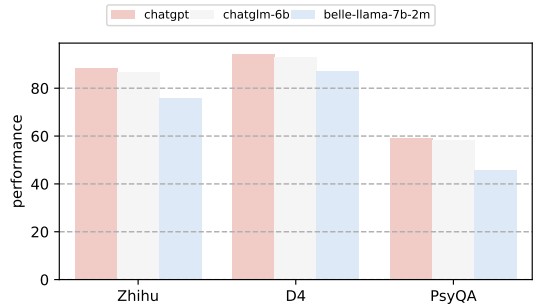
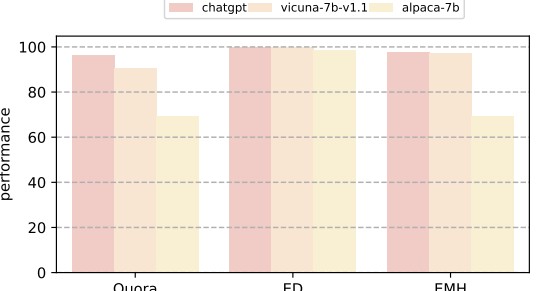

Figure 8: The win rate of responses generated by *M-Cue* CoT compared with the ground truth on **three Chinese** datasets (left) and **three English** datasets (right) in terms of **acceptability**, including several state-of-the-art LLMs.

performs much better but still lags a little behind by ChatGPT. Due to the longest context in the D4 dataset, we found the former two LLMs tend to confuse their own dialogue role and give general responses, resulting in poor performance. For example, *"I am the system/chatbot"* or *"welcome to my chatroom"*, and *"What can I help you?"* often appears in the responses. For the English LLMs, Vicuna achieves comparable performance with ChatGPT in every dataset, and even better in EMH, leading the Alpaca by a noticeable margin. In addition, we can see that the ED dataset is relatively easy to beat since all English LLMs reach almost 100% win rate even though the maximum context length of Alpaca is only 512. Anyway, we conclude that our method is capable of generating more helpful responses than the ground truth, considering the different aspects of user statuses.

Thirdly, we emphasize the performance gap when comparing the ground truth responses is small between LLMs, especially for English LLMs. The Vicuna and ChatGPT achieve almost the same win rate at both ED and EMH datasets. Besides that, putting Figure 7, 8 with Table 2, 3 together, it can be found that the win rate of our method compared with ground truth is relatively higher than

compared with standard prompting, revealing the strong capability of LLMs again. Since our main focus is to prove our method is better than standard prompting instead of ground truth response, we use standard prompting as the baseline during our main experiments.

## C Discussion

In this section, we discuss two key problems: the evaluation of LLMs and the path to more powerful LLMs.

**Illusion of evaluation.** Putting Figure 7 and Figure 5 together, it is plausible to reach two contradicting conclusions about the performance of different LLMs: 1) CHATGPT > CHATGLM-6B > BELLE-LLAMA-7B-2M from Figure 7; and 2) CHATGLM-6B > CHATGPT > BELLE-LLAMA-7B-2M from Figure 5. Although it seems unreasonable, it is indeed the case when most of the responses generated by CHATGPT and CHATGLM-6B are better than ground truth, and then most of the responses generated by CHATGLM-6B are better than CHATGPT. We emphasize that the number of test samples and the choice of baseline (*i.e.,* compared response) plays a key role in evaluation. If the baseline is too weak or the gap is too small, the win rate of different

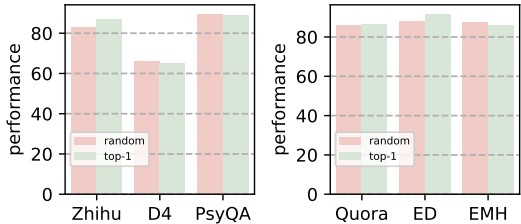

Figure 9: The win rate of responses (helpfulness) generated by `ChatGPT` under different demonstration selection strategies under one-shot setting v.s. responses under the zero-shot setting, using *M-Cue* CoT.

LLMs compared with the baseline may be misleading. The LLM evaluation still is a very difficult problem, and thus we provide different aspects of evaluation to enhance the completeness of our paper.

## D Helpfulness Analysis of Planning Step

| Method | Chinese | | | English | | |
|--------|---------|------|-------|---------|------|------|
| | Zhihu | D4 | PsyQA | Quora | ED | EMH |
| ProcessA | **95.57** | 87.88 | 90.34 | 85.83 | 91.98 | 82.93 |
| ProcessB | 91.18 | 83.57 | 95.13 | 87.67 | **95.35** | 84.82 |
| ProcessC | 92.45 | **88.91** | **95.97** | **89.14** | **96.56** | **84.93** |

Table 7: The win rate of different variants in terms of *helpfulness* with the `ChatGPT` as the backbone.

Table 7 presents the performance of different variants in terms of *helpfulness* and Figure 9 demonstrates the win rate of response of different settings in terms of *helpfulness*. A similar conclusion can be reached as we analyzed in Section 6. We note that the performance of *top-1* selection is relatively lower than *random* selection on PsyQA and EMH datasets in terms of *helpfulness*. We suspect maybe there is a trade-off between helpfulness and acceptability for some specific difficult datasets. We left this into our future work.

**Set of Negative Personas**

用户性格外向，说话大大咧咧。
用户性格比较挑剔，喜欢追问别人。
用户性格忧郁，经常自我怀疑。
用户性格善变，偶尔使用不文明用语。
用户小心谨慎，不愿意相信别人。
用户有些焦虑。
用户心思细腻，优柔寡断。
用户对当前讨论的话题比较敏感。
用户脾气暴躁易怒。
用户内心敏感。
用户性格保守，不愿意接受新事物。
用户很容易会觉得受到冒犯。

- - - - - - - - - - - - - - - - - - - - - - - - - - - - - - - - - - - - - - - - - - - - - - - - - -

The user has an extroverted personality and speaks in a carefree manner.

The user has a critical personality and likes to probe others with questions.

The user has a melancholic personality and often self-doubts.

The user has a fickle personality and occasionally uses inappropriate language.

The user is cautious and reluctant to trust others.

The user is somewhat anxious.

The user is thoughtful and indecisive.

The user is sensitive to the current topic of discussion.

The user has a volatile temper and is easily angered.

The user is emotionally sensitive.

The user has a conservative personality and is unwilling to accept new things.

The user is easily offended.

**Set of Positive Personas**

用户对当前讨论的话题十分好奇，希望系统友善的解答。
用户对当前讨论的话题比较敏感，希望得到支持和鼓励。
用户有较高的要求，追求完美。
用户性格开朗，不拘小节。
用户热情洋溢，待人和善。
用户不歧视他人，充满同情心、爱心。
用户充满对世界的好奇心，善于接受不同的想法。
用户温柔、体贴、乐于交流。
用户脾气平和。
用户很温柔，容忍度高。
用户自尊心很强。

- - - - - - - - - - - - - - - - - - - - - - - - - - - - - - - - - - - - - - - - - - - - - - - - - -

The user is very curious about the current topic of discussion and hopes for a friendly response from the system.

The user is sensitive to the current topic of discussion and hopes for support and encouragement.

The user has high expectations and pursues perfection.

The user has an outgoing personality and doesn't sweat the small stuff.

The user is enthusiastic and treats others kindly.

The user does not discriminate against others and is filled with empathy and compassion.

The user is curious about the world and open to different ideas.

The user is gentle, caring, and enjoys communication.

The user has a calm temperament.

The user is very gentle and has a high level of tolerance.

The user has a strong sense of self-esteem.

Table 8: Some collected personality of users.

Table 9: The prompt to collect the personality of users exhibited in the dialogue.

| Dialogue | User: Is there anything you think you should know about having a baby, but no one tells you? |
|---|---|
| | System: 1. Precautions during pregnancy: such as maintaining good eating habits and reducing alcohol consumption, ... |
| | 2. Possible emergencies during childbirth: ... |
| | User: Is there anything that can help me have a baby successfully? |
| Step-1 | Please infer the user status .... |
| Response | Based on the conversation history, it can be inferred that the user is a detail-oriented and thoughtful person who cares more about personal health and children's health. At the same time, it can be seen that the user has certain anxiety and uncertainty before giving birth and needs more information and advice to help her prepare. |
| Step-2 | Based on the context of the conversation and the user status such as ..., what aspects should the system pay attention to when responding? |
| Response | Based on my speculation about the user's status, my reply needs to consider the following aspects: |
| | 1. Dialogue style: According to the user's personality, my reply may need to be more gentle, and meticulous, and reflect understanding and empathy for the user's concerns. |
| | 2. Needs and preferences: According to the preferences and needs of the user's personality, my reply needs to pay more attention to the user's feelings, ... |
| | 3. Emotional state: ... |

Table 10: An example of additional planning step. We highlight the intermediate reasoning results in  blue .