# OpenReview forum: "Cue-CoT: Chain-of-thought Prompting for Responding to In-depth Dialogue Questions with LLMs"
_EMNLP/2023/Conference — EMNLP 2023 Findings_

### Official Review · Reviewer_cvAj · 2023-07-31

**Typos Grammar Style And Presentation Improvements:** 1. l. 74-77
**Soundness:** 3

**Excitement:**

3: Ambivalent: It has merits (e.g., it reports state-of-the-art results, the idea is nice), but there are key weaknesses (e.g., it describes incremental work), and it can significantly benefit from another round of revision. However, I won't object to accepting it if my co-reviewers champion it.

**Missing References:**

Largely complete. Found two more recent works related to Chain-of-Thought prompting for dialogue response generation:
* [Lee et al. (2023)](https://aclanthology.org/2023.findings-acl.277/)
* [Press et al. (2022)](https://arxiv.org/abs/2210.03350)

**Paper Topic And Main Contributions:**

In order to make the response generation of LLM-based dialogue systems more personalized and engaging, this paper presents a modified Chain-of-Thought prompting technique featuring an intermediate reasoning step that finds linguistic cues (e.g., personality, emotion, and psychology) about user statuses in the dialogue.
The authors contribute a Chinese- and English-language benchmark consisting of 6 dialogue datasets and 5 LLM-based dialogue systems and find that both variants of their proposed Cue-CoT method outperform other prompting techniques in automated evaluation metrics as well as helpfulness and acceptability (human evaluation).

**Reasons To Accept:**

* Simple, straightforward extension of CoT for personalized response generation in dialogue systems
* Comprehensive evaluation on both automated metrics and with human annotators
* Well contextualized in related work
	* Choice of datasets is well motivated.

**Reasons To Reject:**

* Imprecise language
	1. l. 245: "the user status is added as intermediate reasoning…". I find this ambitious, since the user status is not added. O-Cue CoT will first generate the user status and then continue generating the response.
	2. l. 392: "in pair-wise manner": Compare the response generated using Cue-CoT with what? The response generated using the standard prompt? This is unclear to the reader until seeing Tables 2 and 3.
* Human evaluation metrics
	* l. 017-018 / l.488-491: It is unclear how the chosen human evaluation metrics are a valid proxy for personalization and engagement (which are stated as the primary motivation in the beginning). The connections to helpfulness and acceptability are not apparent. Therefore, the initial research question is not sufficiently answered by the evaluation setup.
* Misinterpretation and overclaiming of results
	1. l. 446-447: I'm not sure why you specifically compare ChatGLM with BELLE, since except for the case of M-Cue with zero-shot, ChatGLM also performs better than BELLE using M-Cue with the one-shot setting. I wonder if this comparison is really helpful and meaningful. It both confuses and impresses me that this is the only case where ChatGLM performs better than BELLE.
	2. l. 461-462: "ChatGLM and BELLE under the one-shot setting are much better under the zero-shot setting". M-Cue can't make that decision either. On some datasets, e.g. D4 and Zhihu, helpfulness and acceptability ratings are even much higher using zero-shot settings. What's more, since the ratings of ChatGLM with one-shot settings are not reported, it's not convincing to give this conclusion.
	3. l. 498-499: Since $\kappa$ is mostly around 0 to 0.1 and half of accuracies are under 50 (acc above 50%: in table 4: 12/24, in table 5: 9/24; $\kappa$ above 0.1: in table 4: 9/24, in table: 7/24).

**Reproducibility:**

3: Could reproduce the results with some difficulty. The settings of parameters are underspecified or subjectively determined; the training/evaluation data are not widely available.

**Reviewer Confidence:**

4: Quite sure. I tried to check the important points carefully. It's unlikely, though conceivable, that I missed something that should affect my ratings.

---

> ### Author Rebuttal · Authors · 2023-08-28
>
> We greatly appreciate your valuable comments and suggestions. Regarding all of the concerns you raised:
>
> **Reasons to Reject:**
>
> * Imprecise language
>
>     1. For the one-shot experiments, given the dialogue context input $c_*$, we select a demonstration ($c_1 \rightarrow s_1, r_1$) according to $c_*$ to form the final input, which is ($c_1 \rightarrow s_1, r_1;c_*$).
>     Hence in line 245 *``the user status is added as intermediate reasoning results"*, the user status here refers to the user status ($s_1$) stored in the demonstration.
>
>     2. In lines 401 to 409 (soon after line 392), we explain the evaluation method in detail, including the pairwise comparison and the reasons to do it in this way.
>
> * Human evaluation metrics
>
>     1. Human evaluation also uses the same metrics in Table 2 and Table 3 by considering the helpfulness and acceptability **giving the template** in Appendix A.3 with slight modifications. Our primary motivation and human evaluation metrics (helpfulness and acceptability) are closely interconnected. We have carefully chosen testbeds (such as Zhihu, D4, PsyQA, etc.) to assess whether the model can (1) accurately answer users' questions, (2) interact with empathy, and (3) provide mental health support. If the model can understand users' states, including personality, emotions, and psychological conditions, it will be able to better grasp their needs and offer more personalized, engaged responses. As a result, the model will generate knowledgeable and helpful replies (measured by helpfulness), while also making users feel more at ease and engaged throughout the conversation (measured by acceptability). Therefore, helpfulness and acceptability are effective evaluation metrics to determine the model's ability to conduct personalized and engaging conversations across the six selected datasets, with our carefully designed evaluation prompts (shown in Appendix A.3 following the setting of vicuna [4]).
>
>
> * Misinterpretation and overclaiming of results
>
>     1. Sorry for the confusion but we **have not claimed** this is the only case where ChatGLM performs better than BELLE since our full claim is ``We surprisingly found that ChatGLM performs worst out of the three LLMs **using O-Cue but better than BELLE using M-Cue under the zero-shot setting**".
>     As you mentioned ChatGLM performs better than BELLE using M-Cue under both zero-shot and one-shot settings, it also confuses us why it performs worse using O-Cue under zero-shot settings (lines 445 to 446).
>     Hence we carefully check the outputs of each LLM and provide detailed explanations (lines 448 to 455).
>     In conclusion, ChatGLM performs generally better than BELLE, but the complex instructions following capability may need to be improved.
>
>
>     2. We would like to revise this claim to be "ChatGLM and BELLE under the one-shot setting are much better under the zero-shot setting **using standard prompting method**", since we found that the LLMs are more likely to generate general responses (such as *"As an AI language model"* or *``what can I do for you?"*) under zero-shot rather than satisfying (personalized and engaged) responses as under one-shot (e.g., for D4, ChatGLM generates 21 general responses out of 100 dialogues under one-shot, while 67 out of 100 under zero-shot.). We will release all experimental results with the benchmark once accepted.
>
>     3. Firstly, we would like to address certain misconceptions. In Tables 4 and 5, we present the results of assessing response pairs in varying order. Specifically, "S" denotes responses generated by Cue-CoT, while "O" indicates those generated by standard prompting. The designations "S-O" and "O-S" represent the alternative sequencing order of these two subsets of responses. As revealed in Tables 4 and 5, the "S-O" sequence tends to align less efficiently with human annotations compared to "O-S". Therefore, **we decided to utilize the "O-S" structure for organizing responses in our evaluation template in all experiments**.
>
>          **Considering only the "O-S" structure**, the accuracy and Kappa values **are not low** (acc above 50\%: 12/12 in Table 4, 6/12 in Table 5; Kappa above 0.05: 11/12, 10/12). For the relatively low Kappa, we note that it is due to that humans are more likely to award equal scores (0), while the Language Model (LLM) provides more resolute responses (as indicated in footnote 12 in the paper). To further address your concerns, we recruited the original 3 master students to re-evaluate the responses labeled as "tie" after shuffling. The following two tables effectively demonstrate the alignment scores (in terms of accuracy and Kappa) between human evaluations and automatic evaluations by ChatGPT. **It is evident that ChatGPT consistently achieves a minimum accuracy of 64\% and a Kappa score in (0.1, 0.4) under the "O-S" setting. Furthermore, "O-S" is consistently better aligned with human evaluation than "S-O" in all datasets.**
>
>
>     Table 1: The human evaluation results after re-evaluating the tie(0) on Chinese Datasets.
>
>     | Method | Order | Zhihu | D4 | PsyQA |
>     |--------|-------|-------|----|-------|
>     | | | **helpfulness** | | |
>     | | | | | |
>     | O-Cue | S -- O | 64 (0.23) | 42 (0.13) | 44 (0.06) |
>     | | O -- S | 66 (0.37) | 76 (0.36) | 72 (0.17) |
>     | | | | | |
>     | *M-Cue* | S -- O | 45 (0.14) | 67 (0.08) | 37 (0.09) |
>     | | O -- S | 80 (0.23) | 74 (0.28) | 84 (0.18) |
>     | | | | | |
>     | | | **acceptability** | | |
>     | | | | | |
>     | *O-Cue* | S -- O | 60 (0.16) | 56 (0.14) | 46 (0.04) |
>     | | O -- S | 70 (0.44) | 64 (0.23) | 72 (0.46) |
>     | | | | | |
>     | *M-Cue* | S -- O | 51 (0.16) | 69 (0.23) | 64 (0.09) |
>     | | O -- S | 74 (0.18) | 75 (0.25) | 64 (0.12) |
>
>     Table 2: The human evaluation results after re-evaluating the tie(0) on English Datasets.
>
>     | Method | Order | Quora | ED | EMH |
>     |--------|-------|-------|----|-----|
>     | | | **helpfulness** | | |
>     | | | | | |
>     | *O-Cue* | S -- O | 34 (0.08) | 44 (0.15) | 42 (0.05) |
>     | | O -- S | 68 (0.09) | 80 (0.19) | 78 (0.22) |
>     | | | | | |
>     | *M-Cue* | S -- O | 51 (0.18) | 53 (0.17) | 60 (0.30) |
>     | | O -- S | 82 (0.23) | 79 (0.31) | 81 (0.35) |
>     | | | | | |
>     | | | **acceptability** | | |
>     | | | | | |
>     | *O-Cue* | S -- O | 28 (0.05) | 34 (0.09) | 34 (0.08) |
>     | | O -- S | 66 (0.12) | 76 (0.15) | 81 (0.57) |
>     | | | | | |
>     | *M-Cue* | S -- O | 49 (0.13) | 51 (0.15) | 50 (0.21) |
>     | | O -- S | 84 (0.25) | 82 (0.32) | 75 (0.14) |
>
>
> **Missing References:**
>
> We will add these two papers in the revised version.
>
> **Typos:**
>
> 1. At the end of lines 76 to 77, there is a sentence: *" but the later reasons step by step"*. We noticed there is a typo *"later" --> "latter"* and we will fix it.
>
> 2. For the 4th question, please refer to the response above. For other typos: we will revise it accordingly.
>
> Finally, we wish to extend our heartfelt appreciation for your valuable comments and suggestions. With your insights in mind, we are confident that our paper will be further refined with only minor modifications.
>
> [1] Exploring the use of large language models for reference-free text quality evaluation: A preliminary empirical study
>
> [2] Is chatbot a good nlg evaluator? A  preliminary study.
>
> [3] Is chatgpt equipped with emotional dialogue capabilities?
>
> [4] Judging LLM-as-a-judge with MT-Bench and Chatbot Arena

---

### Official Review · Reviewer_U5Qt · 2023-08-02

**Soundness:** 3

**Excitement:**

3: Ambivalent: It has merits (e.g., it reports state-of-the-art results, the idea is nice), but there are key weaknesses (e.g., it describes incremental work), and it can significantly benefit from another round of revision. However, I won't object to accepting it if my co-reviewers champion it.

**Paper Topic And Main Contributions:**

This paper focuses on how to utilize large language models to generate more personalized and effective responses. The specific approach entails introducing an intermediate step of extracting clues during answer generation to enhance the generative capability of the large language model.

The Contributions are that this paper constructs an in-depth dialogue evaluation benchmark considering the personality, emotion, and psychology of users exhibited in the conversation, with the goal of aligning with unique user needs and status and proposes two effective dialogue cots: O-Cue CoT and M-Cue CoT, that enable advanced reasoning and planning based on user statuses. Both the O-Cue CoT and M-Cue CoT approaches outperform standard prompting in generating more helpful and acceptable responses for the users.

**Questions For The Authors:**

The experimental results suggest that a comparison with the standard prompting method should be included.

**Reasons To Accept:**

1.The NLP community can use the in-depth dialogue evaluation
 benchmark and dataset presented in this paper when address such problem.
2.  The NLP community can draw inspiration from the proposed method in this paper, which employs large language models by Cue-COT, to address other NLP problems.

**Reasons To Reject:**

1.The method proposed in this paper is relatively straightforward and lacks substantial innovation.
2. The experimental results and analysis in this paper are not comprehensive enough. For instance, it does not thoroughly examine the occurrence of cue errors or evaluate the impact of cue errors on the final results.

**Reproducibility:**

4: Could mostly reproduce the results, but there may be some variation because of sample variance or minor variations in their interpretation of the protocol or method.

**Reviewer Confidence:**

3: Pretty sure, but there's a chance I missed something. Although I have a good feel for this area in general, I did not carefully check the paper's details, e.g., the math, experimental design, or novelty.

---

> ### Author Rebuttal · Authors · 2023-08-28
>
> We greatly appreciate your valuable suggestions and feedback. Regarding all of the concerns you raised:
>
> **Reasons to Reject**
>
> 1. As mentioned by *reviewer-qL3t* and *reviewer-cvAj*, our method is interesting, and straightforward and could be easily extended to other aspects in different domains. Moreover, we are the first to identify in-depth dialogue scenarios and propose Cue-CoT (also with a carefully designed demonstration selection strategy) to generate responses according to the user’s hidden needs and status, leading to a better alignment with the user's expectations and preferences. Our Cue-CoT brings more controllability and interpretability and our extensive experiments demonstrate that our proposed Cue-CoT (O-Cue and M-Cue) outperforms standard prompting methods in terms of both helpfulness and acceptability on all datasets. **We humbly consider our method with our proposed benchmark to be substantial for the community to deepen the understanding of personalized dialogue generation (also supported by reviewer qL3t) and address other NLP problems by drawing inspiration from our paper (as mentioned by yourself in reasons to accept).**
>
>
> 2. In fact, we have considered evaluating intermediate reasoning results, such as the occurrence of cue errors or their impact on final responses. We discussed the reasons for not evaluating these intermediate results in the Limitations section.
> Specifically, there are two main reasons:
> (1)**The one-to-many problem leads to an explosion of intermediate candidates.**
> When an LLM solves a complex math problem, it can arrive at the final answer through various solutions. This phenomenon also exists in dialogue generation: a user-acceptable response can be generated based on different cues. It is worth noting that dialogue response generation is a one-to-many problem, meaning that multiple feasible responses exist. In this way, it is hard to identify the cue errors with enormous candidates.
> (2)**Incorrect reasoning does not mean a wrong answer.**
> Despite being counterintuitive, many previous works [1,2] found that LLMs regularly use incorrect reasoning to reach the correct final answer at question-answering tasks. Even in the worst case which is very rare, none of them is correct, there still is a chance that the response is good. Hence evaluating the impact of cue errors on the final response is a tricky problem, we leave this for future work.
> Hence it is difficult to determine the impact of different types of cue errors on the final responses. Based on these considerations, we have decided to **directly evaluate the quality of responses**, as has been done in previous CoT studies [3,4].
>
>
>
> **Questions for Authors:**
>
> As stated in lines 401 to 409, we conduct our experiments by comparing response pairs generated by our proposed **Cue-CoT** (a.k.a, One-Cue, and M-Cue) method with the **standard prompting** method respectively. Tables 2 and 3 show the **comparison results (win rate)** between O-Cue (M-Cue) with standard prompting  (91/96 wins in Table 2; 90/96 wins in Table 3). We extensively conduct experiments and explain the setting and methods of evaluation in lines 386 to 410. We also provide corresponding analysis in Appendix B and C.
>
>
> [1] Star: Bootstrapping reasoning with reasoning. NeurIPS 2022
>
> [2] Selection-inference: Exploiting large language models for interpretable logical reasoning. DeepMind 2022
>
> [3] Chain-of-Thought Prompting Elicits Reasoning in Large Language Models. NeurIPS 2022
>
> [4] Automatic Chain of Thought Prompting in Large Language Models

---

### Official Review · Reviewer_qL3t · 2023-08-05

**Soundness:** 3

**Excitement:**

4: Strong: This paper deepens the understanding of some phenomenon or lowers the barriers to an existing research direction.

**Paper Topic And Main Contributions:**

This work proposes a new chain-of-thoughts (CoT) strategy that looks for the linguistic cues in context. Authors conducted extensive evaluations on 6 datasets with 2 language (English and Chinese), and show that linguistic cues about personality, emotion, and psychology could help significantly improve the following inference output in terms of helpfulness and acceptance.

**Questions For The Authors:**

1. Please see my question in reasons to reject.
2. The proposed method claims to have better responding to in-depth dialogue questions. However, it's not clear in the paper about what are in-depth questions. Could you explain more about it?

**Reasons To Accept:**

The main reasons to accept:

1. querying cues from context is an interesting way of CoT, and the methods proposed in this work covers three aspects (emotion, personality, psychology). It seems that this method could be easily extended to other aspects when needed in different domains.

2. Paper is well written and presented clearly. Very easy to follow.

**Reasons To Reject:**

The main reasons to reject:


1. One thing that is missing is: it's not clear to me how well the proposed CoT method is compared to standard prompting. Tables show superior performance of M-Cue when comparing it to O-Cue, but it's not clear how the standard prompting is.

**Reproducibility:**

4: Could mostly reproduce the results, but there may be some variation because of sample variance or minor variations in their interpretation of the protocol or method.

**Reviewer Confidence:**

3: Pretty sure, but there's a chance I missed something. Although I have a good feel for this area in general, I did not carefully check the paper's details, e.g., the math, experimental design, or novelty.

---

> ### Author Rebuttal · Authors · 2023-08-28
>
> Thank you for your valuable feedback and suggestions. For your raised concerns, we will address them one by one as shown in the following:
>
> **Reasons to Reject:**
>
> As illustrated in lines 401 to 409, we directly regard the responses generated by the **standard prompting** method as the baseline and compare them with responses generated by our proposed **Cue-CoT** (a.k.a, One-Cue, and M-Cue) respectively. We argue this setting is more challenging and straightforward since we found that it is relatively easy for these three methods (*standard prompting, O-Cue, M-Cue*) to beat ground-truth responses as shown in Appendix C.1. In this way, Tables 2 and 3 show the comparison results between O-Cue (and M-Cue) with standard prompting methods. Generally, the win rates exceeding 50\% indicate the O-Cue (or M-Cue) is better than the **standard prompting** method (91/96 wins in Table 2; 90/96 wins in Table 3).   We also provide a relative comparison of these three methods as illustrated in lines 396 to 401.
>
> **Questions for the Authors**
>
> 1. Please refer to the above responses.
>
> 2. By "in-depth dialogues," we refer to conversations that encompass a rich array of linguistic cues.
> These cues not only convey the semantic content but also subtly imply details about the user's psychology, emotion, personality, or other pertinent information, such as the system's values or biases.
> These cues, embedded within the dialogue context, are what define the essence of in-depth dialogues. Specifically, we target three major linguistic cues (personality, emotion, and psychological states of the user) in the paper. We briefly introduce in-depth dialogue in lines 41 to 54 and we will add more explanation in the revised version.

---

### Meta-Review · Area_Chair_KM8R · 2023-09-11

**Recommendation:** 3

**Metareview:**

This paper introduces the Cue-CoT method, which uses linguistic cues to improve LLM's responses by adding an intermediate reasoning step. The method searches for cues in dialogue to create more personalized and engaging responses. The authors created a new benchmark using 6 datasets in both Chinese and English, focusing on three linguistic cues: personality, emotion, and psychology. They tested their method with 5 LLMs in both zero-shot and one-shot scenarios. Results show that Cue-CoT performs better than standard prompting methods in terms of helpfulness and acceptability.

Reviewers gave soundness scores of (3, 3, 3). All reviewers found the paper well-written, and post-rebuttal, agreed that the evaluations were thorough, considering both automatic metrics and human assessment.

Excitement scores were (4, 3, 3). The overall excitement is moderate due to some overstatements in the results, unclear writing, and an incremental rather than transformative methodology.

---

### Decision · Program_Chairs · 2023-10-07

**Decision:**

Accept-Findings

**Comment:**

This paper introduces the Cue-CoT method, which uses linguistic cues to improve LLM's responses by adding an intermediate reasoning step. The method searches for cues in dialogue to create more personalized and engaging responses. The authors created a new benchmark using 6 datasets in both Chinese and English, focusing on three linguistic cues: personality, emotion, and psychology. They tested their method with 5 LLMs in both zero-shot and one-shot scenarios. Results show that Cue-CoT performs better than standard prompting methods in terms of helpfulness and acceptability.

Reviewers gave soundness scores of (3, 3, 3). All reviewers found the paper well-written, and post-rebuttal, agreed that the evaluations were thorough, considering both automatic metrics and human assessment.

Excitement scores were (4, 3, 3). The overall excitement is moderate due to some overstatements in the results, unclear writing, and an incremental rather than transformative methodology.